# CCS Projects: How Regulatory Framework Influences Their Deployment

**Natalia Romasheva and Alina Ilinova *** 

Organization and Management Department, Saint-Petersburg Mining University,
Saint-Petersburg 199106, Russia; Smirnova_NV@pers.spmi.ru
***** Correspondence: Ilinova_AA@pers.spmi.ru; Tel.: +7-921-349-3472

**Abstract:** Preventing the effects of climate change is one of the most pressing challenges of this century. Carbon capture and storage (CCS) technology takes up a promising position in the achievement of a low-carbon future. Currently, CCS projects are implemented not only for $CO_2$ storage but also for its usage in industries, in conformity with the principles of a circular economy. To date, a number of countries have accumulated experience in launching and implementing CCS projects. At the same time, the peculiarities and pace of technology development around the world remain different. This paper attempts to identify key factors that, first, generally affect CCS projects deployment, and second, create favorable conditions for CCS technologies development. Based on an extensive literature review and the experience of different countries, classification and interpretation of these factors are offered, justifying their impact on CCS projects. As a result of this paper, the authors present an assessment of the maturity of policy incentives and regulations in the field of CCS for different countries with revealed dependence between the level and effectiveness of CCS projects' implementation, confirming the adequacy of the offered approaches and identifying measures that ensure success in CCS. The methodology of this study includes case studies, a modified PEST analysis, system-oriented analysis, the checklist method, and regression analyses.

**Keywords:** CCS projects; carbon dioxide; capture and storage; regulatory framework; policy incentives; government regulation; factors; assessment; CCS deployment; circular economy

## 1. Introduction

According to the BP Statistical Review of World Energy (2019) [1], the volume of the global $CO_2$ emissions was 33.9 billion tons in 2018, which was 2% higher than in 2017 (the fastest growth rate in seven years) [1]. Russia is one of the largest sources of greenhouse gas emission in the world (1.55 billion tons in 2018), after China (27.8% of global $CO_2$ emissions), the US (15.1%), and India (7.2%) with a share of around 4.5% [1].

One of the promising ways of reducing greenhouse gas emission is carbon capture and storage (CCS) technology, which includes the capturing of the gas from the industrial sources, its compression and transportation, sometimes its usage in the industrial purposes and the long-term underground storage. The efficiency and safety of the technology is proved by successful realization of the number of CCS and carbon capture, utilization, and storage (CCUS) projects around the world.

The most typical type of CCUS projects are the enhanced oil recovery (EOR)-$CO_2$ projects directed to the increase in oil recovery. However, CCUS projects are now implemented not only in the oil and gas industry but also in the coal industry (in particular, at coal power plants), in the production of steel and ethanol, gas preparation, the production of mineral fertilizers, etc. This technology can have a potentially perspective use in the cement and chemical industries [2], as well as for the production of durable carbon materials [3].

In the research, the authors do not divide CCS projects into these two groups (or more [4]), implying $CO_2$ capture and storage projects in general as CCS. CCUS projects present a circular system where the emissions are minimized and carbon dioxide is used in the industry or is recycled into useful products facilitating the transition to a circular economy [5–7]. These projects support an industrial economy that relies on the "restorative capacity of natural resources" [8] and aims to minimize emissions. The authors of reference [5] reported that a more circular economy can make deep cuts to emissions from heavy industry; in an ambitious scenario, as much as 3.6 billion tons per year globally. The transition to a circular economy offers economic opportunities for business and science, health and safety, as well as cutting $CO_2$ emissions.

According to the Global CCS Institute [9], the global portfolio of large-scale CCS projects has expanded to 43 in 2019; 18 of CCS facilities are in operation and five—under construction, mostly in North America. There are 20 CCS projects at early and advanced stages of development, mostly in China and Europe [9].

Currently, there are no CCS projects running in Russia; however, according to experts, Russia has a huge potential for their implementation because of the vast volume of $CO_2$ emission and existence of geological storage for $CO_2$.

The present rate of CCS projects development does not correspond to the rate of increase in greenhouse gas emissions leading to climate change. CCS projects are outside of the direct economic benefits and relate mainly to environmental outcomes; they are extremely expensive and technologically difficult.

In most cases, CCS projects are run by the state or are implemented with significant financial support from the government; they raise safety issues and govern the interests of a rather wide range of stakeholders [10], such as the state, business, society, etc.

In such countries as the US, Canada, Great Britain, Norway, Australia, and China, the government plays an important role in the development of CCS projects. The listed countries invest in support of this activity and systemically improve the policy in this field. As an example, Australia's captured and injected $CO_2$ volume was about 200,000 tons in 1998–2017, which resulted in expenses of more than 1 billion Australian dollars [11].

The lack of adequate support for CCS has contributed to the relatively slow pace of project deployment to date in many countries. Therefore, this paper attempts to explore key factors that create favorable conditions for the development of CCS technologies in different countries with a systematization of these factors, identification of their manageability, assessment of the level of countries' development in that respect, and recommendations for the countries that lag behind in CCS. Moreover, the key drivers for political "backing" of CCS were identified.

A great number of researchers deal with CCS technologies and projects in various context. Previous studies examine the role and challenges of political support in deploying large-scale CCS projects [12,13], a selection of country case studies [12], barriers to CCS development [14–16], the roles and responsibilities of stakeholders in CCS projects [17,18] and more. Several publications are also devoted to the study of pro and con arguments in CCS projects [19,20].

Most of the analysis has been case studies, often focusing on a particular tool, such as $CO_2$ tax schemes [21,22], and often within a particular country [23,24]. For example, one study compared the various approaches that the United States' state and federal governments have applied to provide regulatory frameworks to address carbon sequestration [25].

In a number of papers, the authors tried to analyze legal and regulatory aspects of a particular part of technological chain of CCS, especially of $CO_2$ geological storage [26,27].

A number of authors investigated the importance of regulatory drivers in the development of CCS projects and diffusion of CCS technologies. All of them concluded that these drivers are crucial, confirming their conclusions with the analysis of different countries' experience [28,29] and CCS project business models [30]. Some authors noted fluctuating support for CCS globally [12], based on the analysis of the International Energy Agency [31].

The existing literature on CCS examines various policy approaches that might stimulate CCS development. For instance, Torvanger and Meadowcroft create a checklist of factors to be considered when allocating government support for low carbon emission energy [32]; von Stechow analyses different CCS support schemes; and Henriksen focuses on challenges to the political and legal framework of CCS [18].

In general, it can be concluded that the influence of various factors on CCS projects, as well as government regulation in this field, is widely considered in the literature. Thus, despite the conducted studies on CCS, this area remains poorly studied in relation to three issues—first, a comprehensive approach to addressing and justifying the full range of factors influencing the implementation of CCS initiatives; second, a formalization of these factors and the creation of a system to assess the maturity of countries' policy and regulatory framework; and third, the situation in Russia.

Considering the differences in approaches to determining the factors influencing CCS initiatives, the present study classifies a range of factors, identifying their manageability, and verifying their impact on CCS. One of the goals of the current study is to learn more about the level of policy and regulatory framework's development in different countries through a checklist and to make recommendations to accelerate the pace of CCS technologies' development.

The novelty of the study involves three key points presented below. First, this paper presents a comprehensive list of factors that positively or negatively affect CCS projects, along with their classification, interpretation, and determination of their manageability. Second, we propose a novel approach to the assessment of maturity of countries' policy incentives and regulatory framework for successful implementation of CCS projects, based on the classification of the factors and consisting of the checklist and an evaluation matrix and correlation analysis. Third, we develop recommendations based on the conducted research, organized in five key areas, aimed at encouraging CCS projects.

## 2. Materials and Methods

The research hypothesis is based on the suggestion that, despite the importance of comprehensive policies for market development, policy incentives and regulatory framework are crucial to the widespread deployment of CCS projects in the world, including Russia. Considering this, this paper reports the key findings and conclusions of an analysis of a large number of factors affecting the implementation and distribution of CCS projects, as well as legal and political incentives in different countries. The authors attempt to give general recommendations for the implementation and development of CCS projects in countries lacking experience in this sphere, including Russia, by examining dozens of such projects that have been in operation, under construction, completed, and cancelled, as well as various policy approaches that might stimulate CCS development. The proposed structure of the research is presented in Figure 1.

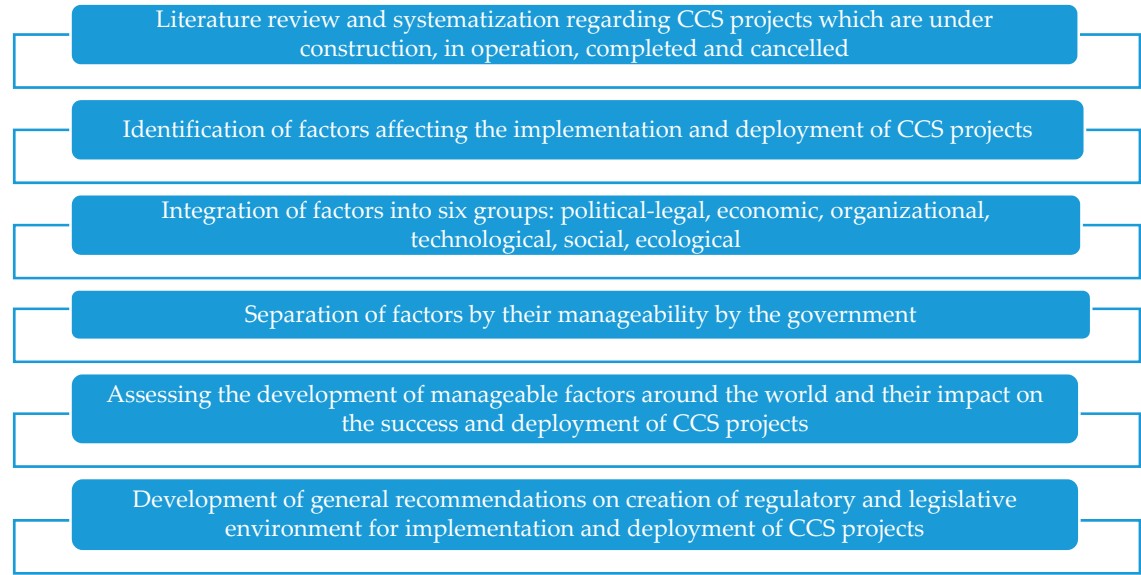

**Figure 1.** The structure of the research.

At the first stage of the research, a desk study was carried out to generalize, analyze, and systematize information on CCS projects in operation, under construction, completed, and cancelled in order to understand the situation around CCS projects in the world, their importance for $CO_2$ emission reduction, and to formulate the detail, scope, and methodology of the follow-up research.

The necessary information was received from available sources such as

1. Companies and CCS projects' websites (for example, see references [33–36]);
2. CCS projects' databases (global CCS institute database [37], the National Energy Technology Laboratory's (NETL) database [38], database provided by the carbon capture and sequestration technologies at MIT) [39];
3. Reports, outlooks, statistics, and data of organizations, institutes, agencies, and official structures that operate to ensure reliable, affordable, and clean energy (The Global CCS Institute [40], the World Energy Council [41], the International Energy Agency [42–44], The Intergovernmental Panel on Climate Change [45], the Carbon Capture and Storage Association [46]).

According to the NETL database, there are more than 300 CCS projects implemented in more than 30 countries worldwide. However, about 25 percent of these projects are frozen, more than 20 percent of projects have been cancelled as a result of the management decision, and about 10 percent of projects are suspended. Moreover, there are 43 large-scale CCS projects at different stages of development operating globally, but most of them are located in North America, so the goal of the European Union of having about 20 operating CCS projects by 2019 has not been achieved.

Therefore, the next stage of the research was to analyze different CCS projects and identify factors affecting their successfulness.

The research included the analysis of various types of projects implemented in all regions of the world, with special attention paid to global projects, and was based on information sources specified earlier, as well as publications and presentations by international experts in scientific electronic and printed journals such as Energies, Energy Procedia, Resources, Applied Energy, International Journal of Greenhouse Gas Control, etc.

To develop an accurate survey, the authors reviewed the literature and interviewed representatives from Russian and foreign mining and energy companies and government authorities who specialize in issues related to energy savings and emissions reductions. Their suggestions and comments helped to create a list of 40 factors affecting the implementation and deployment of CCS projects in the world.

Then, PLEOTES analysis (analysis of political-legal, economic, organizational, technological, ecological and social factors) (based on advanced version of classical PEST analysis) was employed, which was modified by adding one more group of factors that allowed for a clearly divide list of political-legal, economic, organizational, technological, social, and ecological factors (Figure 2). In this methodology, we considered the factors that reflect both the impact on the projects' development and the impact of projects on the external environment.

Then, the factors were divided into manageable, partly manageable, and not manageable. The authors argued that the development of CCS project depends firstly on manageable and partly manageable factors that represent or can be regulated by political and legal framework. Therefore, policy and legal incentives on the case of CCS projects in different countries were examined to identify key political drivers and their effectiveness.

A checklist method was proposed to assess the maturity of countries' policy incentives and regulatory framework in the field of CCS technologies and projects deployment. The authors have developed a list of research questions correlated with 19 political-legal manageable factors presented in Table 1 Each question has three possible answers with different scores, the total amount of which determines, according to authors' assumptions, the level of development (maturity) of regulatory framework. First, the authors suggest "yes" or "no" answers for each question; then, a clarifying answer with different scores is possible. The purpose of this checklist is to provide an assessment for identification and scoping of level of regulatory framework development in different countries to identify, finally, those measures that are critical for CCS projects implementation. Moreover, quantitative assessment permits a quantitative comparison between different countries and situations.

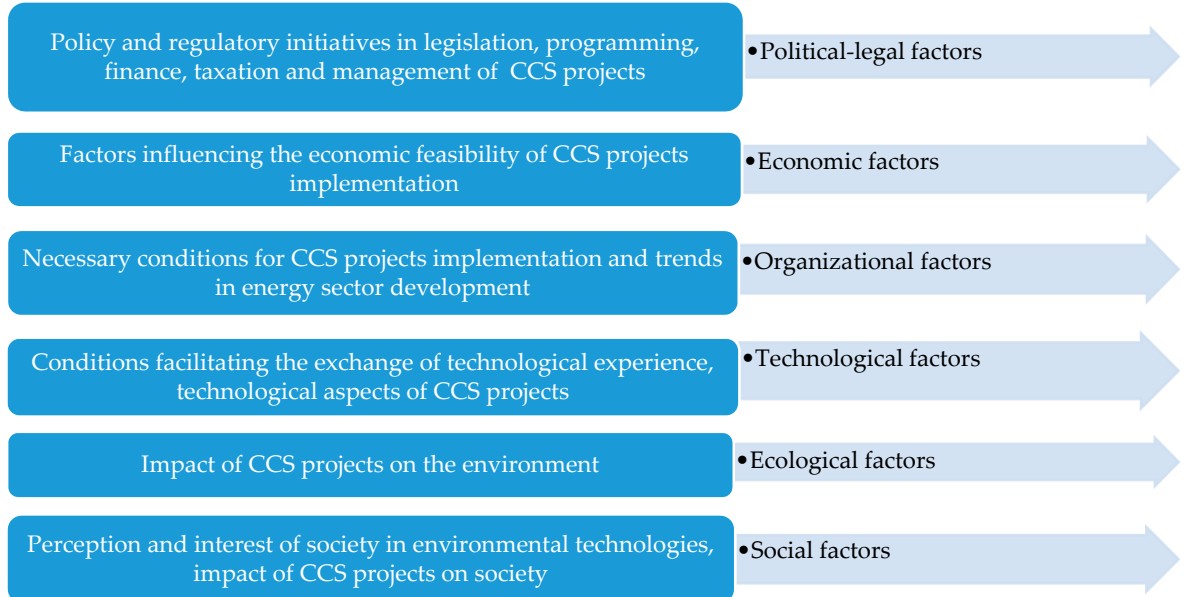

**Figure 2.** PLEOTES analysis.

Finally, correlation analysis was used to verify the accuracy of the assessment of the countries' policy incentives and regulatory framework maturity.

## 3. Results and Discussion

### 3.1. Factors Affecting CCS Projects

Table 1 presents a classification of the factors influencing CCS projects deployment.

**Table 1.** Factors affecting the implementation and deployment of carbon capture and storage (CCS) projects [12–14,17,20,24–30,32,47–49].

| Group of Factors | Factors | Manageability * | | |
|---|---|---|---|---|
| | | 1 | 2 | 3 |
| 1. Political-legal factors | 1.1 Kyoto protocol and Paris climate agreement (ratification, withdrawal) | V | | |
| | 1.2 Climate and energy policies of the country | V | | |
| | 1.3 Government programs, strategies for implementation of CCS projects, $CO_2$ emission reduction roadmaps | V | | |
| | 1.4 Detailed CCS specific laws | V | | |
| | 1.5 Environmental legislation (environmental protection, water, air and waste quality acts) | V | | |
| | 1.6 Standard, limiting $CO_2$ concentration in gas | V | | |
| | 1.7 $CO_2$ tax | V | | |
| | 1.8 Tax preferences for companies implementing CCS projects | V | | |
| | 1.9 Carbon capture tax credit | V | | |
| | 1.10 Emission trade scheme | V | | |
| | 1.11 Direct financial support for CCS projects implementation by different state funds and structures | V | | |
| | 1.12 Government support for R&D research | V | | |
| | 1.13 International cooperation on CCS projects | V | | |
| | 1.14 $CO_2$ storage permitting process | V | | |
| | 1.15 CCS technology promotion institutes and organizations | V | | |
| | 1.16 Predictable legal framework | V | | |
| | 1.17 Promoting environmentally responsible business | V | | |
| | 1.18 Educational tools at all levels of education | V | | |
| 2. Economic factors | 2.1 Oil prices | | V | |
| | 2.2 Capital costs of CCS projects | | V | |
| | 2.3 Commercial efficiency of CCS technological schemes | | V | |
| | 2.4 Cost of energy required for CCS projects | | V | |
| | 2.5 $CO_2$ prices | | V | |
| | 2.6 Private financing of CCS technologies | | | V |
| 3. Organizational factors | 3.1 Level of economic development of the country | | | |
| | 3.2 Focus on traditional energy sources | | | |
| | 3.3 Renewable energy usage | | V | |
| | 3.4 Presence of the saline aquifers for $CO_2$ storage close to the sources of emissions | | V | V |
| | 3.5 Presence of the depleted hydrocarbon reservoirs close to the sources of emissions | | | V |
| 4. Technological factors | 4.1 Advanced centers for CCS technologies promotion (development and implementation) | | V | |
| | 4.2 Development and implementation of other environmental technologies | | V | |
| | 4.3 Non-maturity of technologies used in different stages of CCS chain | | V | |
| 5. Ecological factors | 5.1 Significant $CO_2$ emissions | | V | |
| | 5.2 Possibility of $CO_2$ leaks from geological formations | | V | |
| 6. Social factors | 6.1 Public acceptance of CCS projects | | V | |
| | 6.2 Non-commercial organizations' attitude to the $CO_2$ storage | | V | |
| | 6.3 Employment deficit in the region | | V | |
| | 6.4 Public interest in other environmental technologies | | V | |
| | 6.5 Impact on economic activity by locals (farming, agriculture, fishery) | | V | |
| | 6.6 Monetary burden on taxpayers | | V | |

* 1—it means that the factor can be managed and controlled by various state authorities; 2—it means that state authorities can partly manage and control factors by creating special condition primarily reflecting the factors of group 1; 3—it means that factors can't be managed and controlled by various state authorities.

We have identified the manageability of these factors and concluded that the first group of them are manageable by different government structures and could be connected with the other groups, and particularly all factors, positively or negatively affecting them.

We realize that there are different law systems in different countries, but political-legal factors are managed and controlled by the state authorities in all countries in the world, while other factors can be partly managed and controlled depending on legal possibilities. However, upon further analysis, we consider and compare only political-legal factors.

We have tried to formalize the representation of the revealed factors; however, we are aware that some assumptions were made concerning its group affiliation and interrelations between them.

We present full list of factors that can influence CCS projects' implementation and deployment, but these factors are not detailed depending on type of project, and we assume that a set of factors from this list can be limited and specific for each project. For example, oil prices and the presence of depleted hydrocarbon reservoirs close to the sources of emissions can be considered as crucial factors in cases of CCUS (EOR-$CO_2$) project commercialization and do not affect the implementation of other types of CCS projects.

Appendix A provides a fairly detailed description of the factors with a detailed justification for their significance and meaning, as well as their influence on CCS activities. It is obvious that each factor is very important for CCS implementation and deployment, but each has a different value.

According to our approach, political-legal factors can be managed and controlled by various state authorities and influence other factor group, so they are crucial for the development and deployment of CCS project.

### 3.2. Assessment of the Maturity of Countries' Policy Incentives and Regulatory Framework in CCS

This paper attempts to evaluate the importance of political-legal factors through the assessment of the maturity of the countries' incentives policy and regulatory framework according to proposed checklist (Appendix A). For the assessment, seven countries (the US, the UK, Australia, China, Germany, Canada, and Norway) were chosen for the following reasons:

1.  All these countries play a significant role in $CO_2$ emissions, have a sufficiently high level of economic development, and have experience in CCS project implementation.
2.  $CO_2$ storage is allowed in these countries. For example, in Estonia, Finland, and Ireland, $CO_2$ storage is permanently prohibited, except for research purposes. In Latvia, Austria, the Czech Republic, Poland, and Sweden, $CO_2$ storage was/is temporarily prohibited in order to understand the results of demonstration projects or until the large-scale deployment of $CO_2$ storage technology [50].
3.  All these countries have huge $CO_2$ storage capacity. For example, Finland and Estonia have no $CO_2$ storage capacity.

The results of the countries' policy incentives and regulatory framework assessment are presented in the Table 2. The total score shows the development and variety of the political-legal environment. The higher the score, the more adequate and suitable legislation base is in the evaluated country is for CCS project implementation. The received results correlates with the CCS projects currently deployed worldwide.

The greatest success in the implementation of CCS projects was achieved by the US, due to the existence of developed, rather stable, but strict and flexible environmental legislation; well-functioning funding; technology-promoting institutes and organizations; different tax-credit mechanism, and the active promotion of CCS technologies to the public. During the last two decades, a significant number of regulations in the United States have been adopted at federal and state levels that promote CCS project implementation and development. Also, a significant number of support funds have been created there; many demonstration and commercial projects have been implemented.

Significant success in the CCS project implementation was also achieved by Canada. Canada's federal and provincial (especially Alberta) legislation has been amended to ensure that the necessary regulatory framework is in place before full-scale CCS projects begin. There is a program in Alberta that provides reduced royalty rates for CCS projects. A technology support fund is also operating

in Canada, as well as a Clean Energy Fund, from which partial financing of ongoing CCS projects takes place.

**Table 2.** Assessment matrix.

| Policy Incentives and Regulatory Framework | Countries | | | | | | |
|:---:|:---:|:---:|:---:|:---:|:---:|:---:|:---:|
| | USA | Canada | Australia | Norway | Germany | Great Britain | China |
| 1.1 | 1 | 2 | 2 | 2 | 2 | 2 | 2 |
| 1.2 | 2 | 2 | 2 | 2 | 2 | 1 | 2 |
| 1.3 | 2 | 2 | 2 | 2 | 1 | 1 | 1 |
| 1.4 | 2 | 2 | 2 | 1 | 1 | 1 | 1 |
| 1.5 | 2 | 2 | 1 | 1 | 1 | 1 | 1 |
| 1.6 | 1 | 1 | 1 | 2 | 0 | 0 | 1 |
| 1.7 | 1 | 1 | 0 | 2 | 0 | 1 | 0 |
| 1.8 | 2 | 2 | 0 | 0 | 0 | 0 | 0 |
| 1.9 | 2 | 0 | 0 | 0 | 0 | 0 | 0 |
| 1.10 | 1 | 1 | 1 | 0 | 0 | 1 | 1 |
| 1.11 | 2 | 2 | 1 | 1 | 1 | 1 | 1 |
| 1.12 | 2 | 2 | 1 | 1 | 2 | 2 | 1 |
| 1.13 | 2 | 2 | 2 | 2 | 1 | 2 | 1 |
| 1.14 | 2 | 1 | 1 | 1 | 1 | 1 | 1 |
| 1.15 | 1 | 1 | 2 | 2 | 1 | 1 | 1 |
| 1.16 | 1 | 1 | 0 | 1 | 0 | 0 | 1 |
| 1.17 | 2 | 2 | 1 | 2 | 1 | 1 | 2 |
| 1.18 | 2 | 1 | 1 | 1 | 2 | 1 | 2 |
| Total score | 30 | 27 | 20 | 23 | 16 | 17 | 19 |

Among European countries, Norway has achieved the greatest success in implementing CCS projects, where the government has adopted a gas energy sales strategy, including $CO_2$ capture and storage.

The political-legal environment in Australia is rather developed and diverse. The Joint Carbon Dioxide Research Center was established in the country, financial support was provided to different demonstration CCS projects, and environmental requirements were tightened. However, Australia's overall score is lower than in the US, Canada, and Norway due to the lack of tax preferences, volatile financing support, and taxation. Some projects were initiated due to the introduction of $CO_2$ tax, which was subsequently canceled and which led to CCS projects being postponed.

Interest in China regarding CCS projects is growing significantly. In China, energy and climate policies, which reflect the goal of reducing carbon dioxide emissions, have been adopted. There are several government-supported CCS project programs. Some grants have been elaborated in the energy budget specifically for CCS, and the government actively tries to raise public cognition. However, China's CCS policies have not been fully reflected in the relevant laws and regulations; there are no tax preferences and tax credit system, and government funding is not enough to implement large-scale CCS projects.

To prove the correlation between the maturity and variety of the political-legal environment and the situation of CCS projects deployment in the world, we compare our results with a number of implemented CCS projects in the considered country. To make an accurate comparison the authors take into account active, as well as, completed, postponed and cancelled CCS projects. The authors introduce an indicator: CCS projects' implementation effectiveness ($E_{CCS}$), which can be calculated by the following formula:

$$E_{CCS} = AC_{ccs}/PCl_{ccs}, \tag{1}$$

$$AC_{ccs} = A_{ccs} + C_{ccs}, \tag{2}$$

where $A_{ccs}$ is the number of active CCS projects, $C_{ccs}$ is the number of completed CCS projects, and $PCl_{ccs}$ is the number of postponed and cancelled CCS projects. Table 3 and Figures 3 and 4 show

that there is a strong correlation between the maturity and variety of the political-legal environment ($D_{pl}$) and the effectiveness of CCS project implementation ($E_{CCS}$).

**Table 3.** Analytical basis for the analysis of the maturity of the political-legal environment and the effectiveness of CCS project implementation.

| Country | CCS Projects [4,37–39] | | | $D_{pl}$ | $E_{CCS}$ |
|---|---|---|---|---|---|
| | Active | Completed | Postponed and Cancelled | | |
| USA | 30 | 49 | 34 | 30 | 2.3 |
| Canada | 7 | 6 | 8 | 27 | 1.6 |
| Australia | 8 | 5 | 11 | 20 | 1.2 |
| Norway | 6 | 0 | 4 | 23 | 1.5 |
| Germany | 1 | 4 | 5 | 16 | 1 |
| Great Britain | 3 | 0 | 6 | 17 | 0.5 |
| China (case 1) * | 7 | 2 | 4 | 19 | 2.25 |
| China (case 2) ** | 5 | 1 | 4 | 19 | 1.5 |

* We consider all types of projects. ** We exclude CCS projects co-financing by Canadian and Australian organizations.

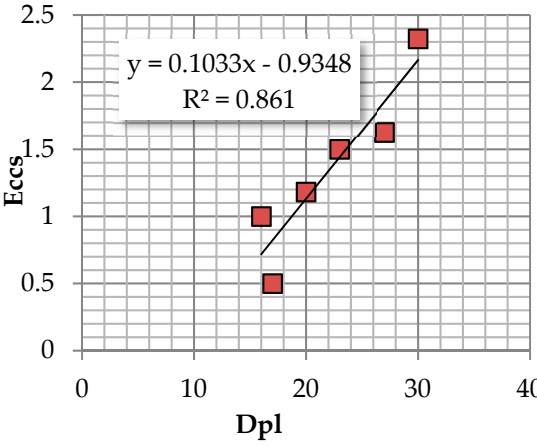

**Figure 3.** Regression analysis of the maturity and variety of political legal environment and CCS projects implementation without China.

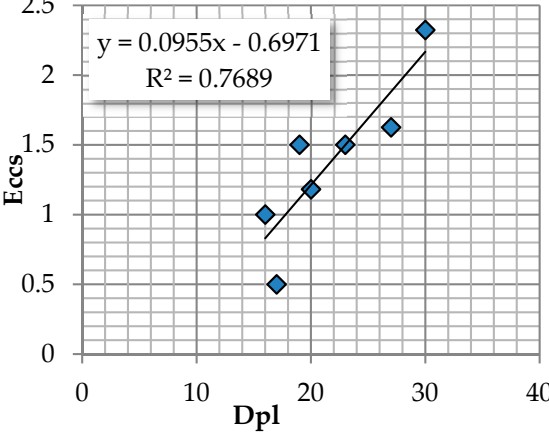

**Figure 4.** Regression analysis of the maturity and variety of political legal environment and CCS projects implementation with China (case 2).

Excluding China (Figure 3), the model is reliable. In Excel, a regression analysis was performed, and it was found that the multiple correlation coefficient is 0.927, the correlation ratio is 0.908, which according to the Cheddock scale indicates a very strong connection between function and the factor. The standard error of the model is 0.258, and the coefficients are 0.472 and 0.02. The absolute

values of the correlation coefficient and correlation ratio are greater than their errors; therefore, the model is adequate. Analysis of variance by the Fisher test (F test) showed that the probability of model inadequacy is 0.007%, and the coefficients are 0.118% and 0.007. This result and further analysis revealed an error in choosing all projects in China for assessment, because several CCS projects in this country were developed with co-financing by Canadian and Australian companies and organizations (for example, Qinshui ECBM Project, Post-Combustion Capture Project in Beijing) with a special government attitude and significant funding. It can also be concluded that, carrying out such an assessment, one must select projects that are comparable in approaches in order to, finally, identify measures that ensure success in CCS.

To get more reliable results, the authors exclude from the analysis CCS projects in China co-financed by Canadian and Australian organizations (Table 3, Figure 4). In Excel, a regression analysis was performed, and it was found that the multiple correlation coefficient is 0.876, which means that the model is also reliable.

The above has also enabled us to validate the developed checklist with the conclusion that it can be used for assessing the level of CCS development for a particular country. Also, identifying the factors that determine success in CCS in general necessitates, as mentioned above, analyzing comparable countries. If the aim is to estimate the level of development of a particular country, the checklist can also be used. In this case the basis for comparison could be above listed countries (as a sample benchmark).

Looking at Table 2 from a methodological point of view, it should be noted that, for highly correlated countries, factors for which the country's score is non-zero are the main determinants of a country's success in implementing CCS initiatives. Using this observation, we were able to formulate general recommendations to establish a policy and regulatory framework that encourages CCS projects' implementation and distribution. These recommendations can be used by countries lacking experience in this sphere, including Russia.

1. Planning and programming. A lot of countries, including Russia, ratified the Paris agreement, so it is important to incorporate the goal of reducing $CO_2$ emissions and integrate CCS initiatives in energy and climate policy. Also, it is necessary to formulate goals and objectives of $CO_2$ emissions reduction and set out a number of actions for the long-term CCS project development in CCS roadmaps and CCS programs or energy strategy. Countries such as the US, Canada, and Norway have made significant progress toward the implementation of CCS technology through a national climate policy, CCS programs, and roadmaps.

2. Financing. It is obvious that low commercial effectiveness, complexity of technologies, high capital, and high operational costs make financial support crucial CCS project implementation. But, accessing to funding is a challenging process adding uncertainty in many countries. For example, about 10 years ago funding programs were announced in the US, Australia, Europe, Canada and Great Britain with $31 billion USD support for CCS projects, but only $3 billion USD was really invested from 2009 to 2014, mainly in Canada and the US. This was the main reason for CCS projects being cancelled in European countries, including Norway and Great Britain. It is therefore important to create stable, balanced, and dynamic financial support programs for CCS projects to provide confidence to business decisions and overcome high initial costs.

3. Legislation. Before a CCS project's implementation, it is necessary to introduce clear legislation for $CO_2$ capture, storage, utilization, and transportation, which should be in accordance with other legitimate activities (for example, with hydrocarbon exploration and production, natural gas storage, drinking water production, geothermal resources development). It should also be predictable but flexible because of the unique characteristics of each project. Lack of special legislation can become a reason to postpone or cancel a CCS project. Also, it is important to create a $CO_2$ storage permitting base. Worldwide experience shows that a permitting process adds uncertainty to project development, and it can be very complicated and time consuming.

4.  Public outreach. To accelerate the deployment of CCS projects, it is important to raise public awareness in CCS technologies, to interact with non-government organization, and implement government programs and guidelines to encourage companies to consider environmentally responsible business as part of their activity. Such measures can influence public perception and lead to positive effects in CCS projects development. A lot of projects were cancelled or postponed due to the negative public attitude toward these technologies.

5.  Taxation and crediting. From our point of view, at the first stage of CCS project's implementation, $CO_2$ tax could not be considered as an effective tool for noncommercial CCS projects.

*3.3. Discussion*

Summarizing the conducted research, the following received results should be mentioned:

- Identification, classification, and interpretation of the factors affecting the CCS projects development;
- Assessment of maturity of policy incentives and regulatory in the field of CCS for different countries through a checklist method and verification of the received results through correlation analysis;
- Identification of the factors and measures creating favorable conditions for CCS technologies development;
- Development of recommendations aimed at establishment a policy and regulatory framework that encourages CCS projects' deployment.

Compared to previous research results in this area [12,15,17,32,51], it can be concluded that the developed system of factors affecting CCS projects is more comprehensive, including an extended list of factors and their interpretation and justification. The proposed approach to the assessment of maturity of policy incentives and regulatory in CCS could replace the descriptive analysis of a country's CCS development in the existing literature [25,26], as well as allows researchers to conduct a quantitative assessment regardless of the degree of a particular country's participation in CCS initiatives. Identification of the factors and measures that create favorable conditions for CCS allows for the development of recommendations to encourage CCS projects.

The main assumptions of the paper are the following:

- Not all factors affect all CCS projects; we have attempted to create a comprehensive list of factors that may influence CCS projects, and it can be limited and adapted, where applicable.
- The importance of the factors was not taken into account in our assessment, although it can vary; we see this as a direction for further research.
- Open sources of information were used, so we assume that some data on country initiatives or CCS projects may be slightly distorted.
- A common approach to assess the level of development (maturity) of policy initiatives and regulatory framework was proposed; it can be applied to any country (even those that are undeveloped in the field of CCS).

In this research, the proposed approach was applied to those countries that are the world leaders in CCS, but this checklist and assessment system may be applicable to any country, including Russia.

## 4. Conclusions

World experience and conducted analysis show that the number of barriers related to economic issues, safety, and public acceptance/resistance for the full-scale deployment of CCS globally is way too high. The absence of direct commercial effects (in most cases) and the importance of such initiatives for carbon dioxide emission reduction determines the peculiarities of CCS projects. In this regard, the role of state regulation for the positive implementation of CCS projects cannot be overestimated.

According to this study, it can be concluded that CCS activities should be permanent, systematic, and balanced; they should be based on other countries' CCS background. CCS projects are local but

are implemented in the context of national and even international interests. So, coordinated efforts to succeed in CCS activities, as well as the participation of all stakeholders (especially government), are needed.

As for Russia, manufacturing plays an extremely important role in the national economy. Many industrial facilities, such as the chemical sector, oil refineries, and other processing industries, produce a huge amount of $CO_2$ (the fourth highest emissions in the world, as noted above). Russia has a significant potential for carbon storage, and its use in $CO_2$-EOR (for example, the Republic of Tatarstan and some regions of Western Siberia have depleted oil fields). In addition, CCUS technologies provide one of the pathways to address the need for a low carbon future and the implementation of circular business models. CCUS technology is necessary to meet circular economy goals, but its widespread deployment will require continued improvements in legal regulations and politics around the world. For the impactful government participation in CCS initiatives, it need to be well informed of all existing barriers, since its support and regulation on CCS is of crucial importance for the implementation of such projects. It includes the support of R&D, cooperation with the business sector and energy industry, as well as the creation of societal acceptability for the full-scale deployment of CCS.

We have come to conclusion that aspects such as the planning and programming of CCS activities at the state level, stable financial support, clear legislation, public outreach, and adequate country-specific taxation and crediting are critical to CCS technologies propagation. Since this sphere is new for Russia, the foundations of state policy in CCS must be based on the recommendations presented above.

**Author Contributions:** Conceptualization, N.R. and A.I.; Formal Analysis, N.R. and A.I.; Investigation, N.R.; Methodology, N.R. and A.I.; Validation, N.R.; Writing—Original Draft Preparation N.R. and A.I.

**Funding:** The research was carried out with the financial support of a grant by the Russian Science Foundation (Project No. 18-18-00210, "Development of assessment methodology of public efficiency of projects devoted to carbon dioxide sequestration").

**Acknowledgments:** The authors are grateful to Amina Chanysheva for her help in editing the paper.

**Conflicts of Interest:** The authors declare no conflict of interest.

## Appendix A

**Table A1.** Interpretation of factors affecting the implementation and deployment of CCS projects [11–32,47–64].

| Factors | Interpretation |
| --- | --- |
| 1.1 | The agreement regulates measures to reduce carbon dioxide in the atmosphere since 2020. The ratification of this agreement contributes to the implementation of CCS projects; withdrawal from the agreement may have a negative impact on the intensification of these projects. |
| 1.2 | The climate and energy policies of states differ significantly in goals and tools. In a number of countries, the course toward reducing greenhouse gas emissions is the main focus of both climate and energy policies, which is reflected in a number of legislative acts, standards, and laws that are adopted on their basis and contribute to the implementation of CCS projects. |
| 1.3 | Government programs, strategies, and $CO_2$ emission reduction roadmaps that are clearly formulated and consistent with strategic and climate policies can encourage CCS project implementation and deployment. However, their inconsistency, isolation, or lack of different factors consideration may become a barrier to the deployment of CCS projects. |
| 1.4 | There are special laws for $CO_2$ transportation, onshore and offshore storage, and estimation of $CO_2$ storage capacity in many countries, which create a clear legal framework and remove the uncertainty for stakeholders. However, in some countries, there are laws that do not allow any storage and create barriers for CCS projects. Also, in some countries, there is a conflict of interests between CCS and other legitimate activities—for example, with hydrocarbon exploration and production, natural gas storage, drinking water, or geothermal resources. Lack of legislation in $CO_2$ storage makes it particularly difficult to deal with unique issues associated with $CO_2$, including licensing authority, monitoring and reporting plans, and corrective and remediation measures. |

**Table A1.** *Cont.*

| Factors | Interpretation |
|---|---|
| 1.5 | Strengthening of environmental legislation, the introduction of standards for pollutants' emissions into the atmosphere, standards for permissible impact on water, and the establishment of requirements for the production and consumption wastes management in order to prevent their harmful impact to the environment—all of these stimulate companies to implement environmental measures, including CCS projects. |
| 1.6 | There are certain requirements for the content of $CO_2$ in the gas for its further transportation or liquefaction, which necessitates its removal during processing, as well as further storage. |
| 1.7 | Carbon tax is a very effective mechanism in cutting back on carbon emissions and its further storage. However, it can be considered to be an aggressive instrument of state policy that negatively influence on companies' commercial efficiency. |
| 1.8 | The government may provide a reduction in taxes (royalties, income tax) for CCS projects, which positively affects to their implementation and distribution. |
| 1.9 | Carbon capture tax credit is a fiscal incentive to support access to the capital for CCS projects. It is a tax credit that can be available to companies that invest in CCS projects, and it is designed to stimulate these projects in all sectors of the economy. However, in a business model in which capture units and $CO_2$ storage enterprises are independent of each other, $CO_2$ subsidy allocation mechanisms need to be established and improved carefully. |
| 1.10 | An emission trade scheme is one of the world's main climate policies to reduce $CO_2$ emissions and can be considered as an effective mechanism to stimulate CCS projects implementation. This mechanism establishes emission reduction commitments for market participants and distributes emission quotas. Participants can buy their quotas to offset excessive emissions or sell their quotas. |
| 1.11 | Financial support from the government is crucial in many cases of the development and diffusion of low carbon technologies. A lot of projects have been supported by different policies, or, in some cases, blends of policies at federal, state, and local levels. However, despite the necessity of financial support, it is important to find the balance between state aid and the industry's own investments in to secure both cost reduction and the efficient performance of projects. |
| 1.12 | There are many R&D research programs all over the world that stimulate the implementation of CCS technologies and are directed to reduce risk and cost while at the same time increasing the available resources in each project. |
| 1.13 | International cooperation on the development and commercialization of new CCS technologies plays an effective role in climate change mitigation. A lot of countries provide funding for CCS projects in cooperation with other countries and through existing programs and institutions. |
| 1.14 | The $CO_2$ storage permitting process is a critical regulatory challenge faced by CCS projects that are common to all regions. A lot of projects all over the world were postponed or cancelled because of a comprehensive $CO_2$ storage permitting process. |
| 1.15 | There are government institutes and organizations that connect parties around the world to solve problems, learn from each other to accelerate the deployment of CCS projects by assisting them, sharing knowledge, and increasing the awareness of the benefits of CCS and the role it plays within a portfolio of low carbon technologies. The creation of different sources of information can be crucial for public awareness and perception of CCS technologies. |
| 1.16 | A predictable legal framework, both national and international, is necessary to deploy CCS. Because of wide range of individual and unique physical conditions in each CCS project, it is important to have a flexible, but predictable, framework that can encourage reduction of $CO_2$ emissions and investment in CCS technologies. |
| 1.17 | There are different government programs and guidelines to encourage companies to consider environmentally responsible business as part of their activity. These help companies understand their influence on the environment and can stimulate the implementation of environmental measures, including CCS projects. |

**Table A1.** *Cont.*

| Factors | Interpretation |
| --- | --- |
| 1.18 | A key part of government policy is to provide students with knowledge, skills, values, and attitudes to realize the importance and role of CCS projects in sustainable development. This policy should be introduced in all aspects of the formal, non-formal, and informal education system in different countries. |
| 2.1 | In cases of CCS-EOR projects, oil prices can be crucial for their commercial efficiency. High oil prices stimulate oil and gas companies to integrate CCS technologies into their activities. |
| 2.2 | Capital cost reduction of CCS projects is complex and crucial for further deployment of CCS. But CCS technologies are still in a learning phase, where costs are harder to control. Experts have identified the high investment cost of CCS as the major challenge preventing the widespread adoption of this technology. It is important to continue government and private support of CCS projects in order to provide cost reduction in the future. |
| 2.3 | Low profitability of CCS is currently among the key barriers for CCS projects. Low profitability is due to combination of increasing operational costs, especially transport and storage costs, and high capital costs. Commercial scale implementation requires a certain level of experience in the technical, operational, and economic feasibility of projects. |
| 2.4 | The cost of energy depends on process plant, capture technology, and storage solutions. The cost of electricity may be much higher when CCS is adopted instead of CCS-EOR for the storage of $CO_2$. The cost of CCS has been previously identified as a major barrier to its adoption. |
| 2.5 | High $CO_2$ prices can encourage participants to apply emission reduction measures comprising the CCS projects. However, significant fluctuations in $CO_2$ prices have been observed recently, which led to an increase in the number of carbon pricing initiatives. |
| 2.6 | Private sector financing of CCS is becoming urgent because of high capital costs and long construction periods. However, the problem is that private investors want to know the first movers' experience and invest money to the approved technology rather than deploy their capital on its development. |
| 3.1 | Due to the high cost and complexity of CCS projects, their implementation is only feasible in countries with a sufficiently high level of development. |
| 3.2 | Fossil fuels will play a significant role in meeting energy needs in the coming decades; in this case, CCS technologies will be in demand to reduce $CO_2$ emissions. |
| 3.3 | Using renewable energy has been a key factor contributing to limiting political support for CCS. Due to limited financial resources, CCS and renewables projects are competing for the same public and private investments. |
| 3.4 | Carbon capture is applicable to different industries (natural gas processing, power generation, iron and steel production, cement manufacturing, etc.), but its storage is expedient if saline aquifers are located near emission sources. Nevertheless, $CO_2$ storage in saline aquifers is considered to be one of the most feasible technologies. |
| 3.5 | Due to huge storage capacity and existing infrastructure, depleted hydrocarbon reservoirs are one of the most favorable storage options, but they must be located near the emission sources. These reservoirs are considered for EOR, which makes them economically more favorable than saline aquifers. |
| 4.1 | Such centers could help in CCS technology development. |
| 4.2 | The lower technical and commercial complexity of renewables' technologies has led to a much greater project success and will be an important factor in competition with CCS technologies, which are considerably more complex than other low emission technologies. |
| 4.3 | Various components of $CO_2$ capture, transport, and geological storage are at different stages of technological maturity. Some capture technologies, particularly in industrial sectors, are commercially efficient, but in the power sector, where the largest potential for CCS deployment resides, full-scale demonstrations remain to be built. |

**Table A1.** *Cont.*

| Factors | Interpretation |
| --- | --- |
| 5.1 | National $CO_2$ emissions are crucial for political decisions on CCS technologies. Most of the research and development activities occur in the states with the highest emissions intensity to prevent negative impact to the environment. |
| 5.2 | The possibility of potential leaks of $CO_2$ is one of the largest barriers to large-scale CCS projects, although well-selected storage sites are likely to retain over 99% of the injected $CO_2$ over 1000 years |
| 6.1 | Public acceptance is key to the deployment of carbon capture and storage locally and globally. There is no general model able to explain public acceptance, but it may include a range of different factors affecting acceptance—attitude, knowledge, experience, trust, perceived costs, risks, and benefits. |
| 6.2 | Negative opinion of different non-commercial organizations often causes projects to be cancelled or postponed. |
| 6.3 | Implementation of CCS projects can create jobs and reduce employment deficits. |
| 6.4 | Interest in other environmentally friendly technologies can provoke negative acceptance of CCS technologies. |
| 6.5 | Implementation of CCS projects may have a negative impact on the activities of the local population. |
| 6.6 | There is an important question: who is going to pay for CCS projects? Non-commercial organizations are against government subsidy for CCS projects. However, if the companies pay for CCS, it increases extra costs and can be the reason of high electricity prices. |

**Table A2.** Checklist for assessing the maturity of countries' policy incentives and regulatory framework.

| Factors | Research Question | Answers and Scores |
| --- | --- | --- |
| 1.1 | Has the country ratified the Kyoto protocol and Paris climate agreement? | No or just in formal way (it means that government does not take this document seriously and implement different initiatives to follow the obligations)—0 points; Yes, but it was decided to withdraw from the document or review the terms of participation—1 point; Yes, not formally—2 points |
| 1.2 | Has the country adopted energy and climate policies and how it reflects the goal of reducing carbon dioxide emissions? | No, there is no climate or energy policy—0 points; Yes, but the goal to reduce $CO_2$ emission is not clearly presented in these documents, or energy and climate policies are just formal documents—1 point; Yes, the course towards reducing greenhouse gas emissions is the focus of these policies and they are not formal—2 points |
| 1.3 | Are there government programs or strategies for implementation of CCS projects, $CO_2$ emission reduction roadmaps, and how are they formulated? | No, there is no government programs, strategies for implementation of CCS projects, $CO_2$ emission reduction roadmaps—0 points; Yes, but they are very complicated and could become barriers for CCS projects' implementation, or they are just formal documents—1 point; Yes, they are clearly formulated and consistent with other documents—2 points |
| 1.4 | Has the government detailed CCS-specific laws, and how do they work? | No, there is no detailed CCS-specific laws—0 points; Yes, but it raises legislation uncertainty—1 point; Yes, it creates a clear legal framework—2 points |
| 1.5 | Does the country have environmental legislation? | No, there is no special environmental legislation—0 points; Yes, there is special environmental legislation, but emission standards cover limited spheres or are rather easy to comply with—1 point; Yes, there is special strict environmental legislation—2 points |

**Table A2.** *Cont.*

| Factors | Research Question | Answers and Scores |
|---|---|---|
| 1.6 | Is gas produced in the country, and are there restrictions on its carbon dioxide content? | No, there is no production in the country—0 points; Yes, there is production, but there is no restriction on its carbon dioxide content, or these restrictions are not significant—1 point; Yes, there is production and restriction—2 points |
| 1.7 | Has the country adopted a $CO_2$ tax and in what amount? | No, there is no $CO_2$ tax in the country—0 points; Yes, there is $CO_2$ tax, but its amount is not enough to push the companies to implement CCS projects—1 point; Yes, there is $CO_2$ tax, and its amount is the reason for the companies to implement CCS projects—2 points |
| 1.8 | Are there tax incentives for companies implementing CCS projects provided in the country? | No, there is no preferences—0 points; Yes, but they are quite limited—1 point; Yes, there is a wide range of preferences—2 points |
| 1.9 | Is there tax credit system for accessing the capital for CCS project implementation provided in the country? | No, there is no tax credit system—0 points; Yes, but it is not well developed—1 point; Yes, there is well developed tax credit system—2 points |
| 1.10 | Is there an emission trade scheme provided in the country and used for CCS project implementation? | No, the emission trade scheme is not used in the country—0 points; No, but it was an attempt to use it or this scheme is under consideration—1 point; Yes, there is well organized emission trade scheme—2 points |
| 1.11 | Is significant financial support organized at different levels in the country, and are they well functioning? | No, there is no any organized financial support or it amount is not enough—0 points; Yes, but it is rather complicated—1 point; Yes, there is well functioning financial support—2 points |
| 1.12 | Is state R&D funding organized in the country? | No, there is no any organized R&D funding—0 points; Yes, but it is rather complicated or not enough—1 point; Yes, there is well functioning system and enough funding—2 points |
| 1.13 | Does the country internationally cooperate on developing and commercializing CCS technologies? | No, there is no cooperation with other countries—0 points; Yes, the country participates in some international programs—1 point; Yes, the country actively provide funding for CCS and CCUS projects in cooperation with other countries—2 points |
| 1.14 | Is the $CO_2$ storage permitting process complicated, and are there any restrictions? | Yes, the $CO_2$ storage permitting process is complicated, takes long time and there are strong restriction for $CO_2$ storage—0 points; Yes, the $CO_2$ storage permitting process is rather complicated, takes time and there are some restriction for $CO_2$ storage—1 point; No, the $CO_2$ storage permitting process is well organized—2 points |
| 1.15 | Is there any special organization or different informational resources in the country to inform the public about CCS technologies? | No, there is no special organization and different resources—0 points; Yes, there are a number of organizations and some information resources, where public can get information about CCS—1 point; Yes, there are a lot of government organizations that hold meetings with public, support informational resources and provide active policy to improve public awareness on CCS technologies—2 points |

**Table A2.** *Cont.*

| Factors | Research Question | Answers and Scores |
|---|---|---|
| 1.16 | Is the legal framework, especially the financing process, predictable or stable? | No, the legal framework is not predictable and always fluctuate—0 point; Yes, the legal framework is predictable, but from time to time it changes—1 point; Yes, there is predictable, but flexible legal framework—2 points |
| 1.17 | Are there different government programs and guidelines to encourage companies to consider environmentally responsible business as part of their activity? | No, there are no programs—0 points; Yes, there are few government programs—1 point; Yes, there are different government programs and guidelines—2 points |
| 1.18 | Are any educational tools used in the country? | No, educational tools are not used in the country—0 points Yes, but rather pointwise—1 points Yes, there are wide range of tools that are used in the country—2 points |

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
