# Peer review of "CCS Projects: How Regulatory Framework Influences Their Deployment"

_resources, doi:10.3390/resources8040181_

Round 1

Reviewer 1 Report

Page 42, Please authors define CCUS.

3.2. Assessment of the maturity of countries’ policy incentives and regulatory framework in CCS

Please describe the implementation of CCS projects in Australia and China as the assessment score in Table 2 shows that the Total score of Australia and China are 20 and 19. It means they are also suitable for CCS project implementation legislation base.

In the case of China, authors could describe projects that include co-financing by Canadian and Australian organizations and the recent projects developed there based on a 'lesson-learned'.

Figure 3. In the China case, what if authors only include CCS implementation which is not co-financed by other countries organizations, will the Eccs of China decrease?

Author Response

Dear Reviewer,

Thank you very much for your efforts, constructive comments and suggestions on our manuscript (ID: resources-654861). The feedback you provided to our manuscript has been helping us to improve and strengthen the paper. We carefully went through your comments and reviewed the manuscript accordingly. Please find below how the issues you raised informed our revision. All changes are in red.

Point 1. Page 42, Please authors define CCUS.

Answer: Thanks a lot for this comment. We put a definition.

Point 2.1 3.2. Assessment of the maturity of countries’ policy incentives and regulatory framework in CCS

Please describe the implementation of CCS projects in Australia and China as the assessment score in Table 2 shows that the Total score of Australia and China are 20 and 19. It means they are also suitable for CCS project implementation legislation base.

Answer: Thank you for your suggestion. We tried to describe the implementation of CCS projects in Australia and China (Lines 268-280).

Point 2.2 In the case of China, authors could describe projects that include co-financing by Canadian and Australian organizations and the recent projects developed there based on a 'lesson-learned'.

Answer: Thanks a lot for your valuable comment. We excluded our idea about «lesson learned» and made some comments on projects that include co-financing by Canadian and Australian organizations (Lines 302-306)

Point 2.3 Figure 3. In the China case, what if authors only include CCS implementation which is not co-financed by other countries organizations, will the Eccs of China decrease?

Answer: Thank you for your valuable suggestion. We included only CCS projects, which were not co-financed by other countries organizations, and the Eccs of China decreased (table 3, figure 4, Lines 311-314)

Reviewer 2 Report

The authors prepared interesting study of the factors influencing CCS projects deployments. However, the manuscript needs some improvements.

1) Line 42: The CCUS should be explained when appears the first time.

2) In the introduction the novelty of the study could be more underlined. 

3) Table 1: I wonder if the conclusions in the table are comparable, because of the different law systems in different countries. For example, in some countries the legal possibilities of price regulation by the goverment are greater then in another countries. 

4) Table 1: It is not clear whether the values of menageability are the averages or medianes (the most frequent) of values of analyzed literature.

5) Line 267: How the Dpl value was evaluated?

6) Figure 3: I suggest other method of correlation evaluation. Simple linear correlation calculation is unsufficient in my opinion. The model should be proved in more appropriated way.

7) Line 286: I have not found the Table 4. Do you mean Fig. 4? 

8) Line 328: In discussion I would add some references to other publiaction (if they have been published so far) treated about the problem.

9) General comment: The authors did not provided collected information about the number of project in several countries, which were analysed. It could provide additional information.

Author Response

Dear Reviewer,

Thank you very much for your comments concerning our manuscript entitled “CCS projects: how regulatory framework influences their deployment” (ID: resources-654861). Those comments are all very helpful for improving our paper. We have studied comments carefully and have addressed all of them. Please see the responds to your comments in the attachment. All changes are in red.

Reviewer 2 

The authors prepared interesting study of the factors influencing CCS projects deployments. However, the manuscript needs some improvements.

Point 1. Line 42: The CCUS should be explained when appears the first time.

Answer: Thank you for your comment, we made it.

Point 2. In the introduction the novelty of the study could be more underlined. 

Answer: Thank you for your suggestion. We tried to underline the novelty of the study at the end of Introduction (Lines 119-126).

Point 3. Table 1: I wonder if the conclusions in the table are comparable, because of the different law systems in different countries. For example, in some countries the legal possibilities of price regulation by the goverment are greater then in another countries. 

Answer: Thank you for your comment. We tried to explain our idea (Lines 211-214).

Point 4. Table 1: It is not clear whether the values of menageability are the averages or medianes (the most frequent) of values of analyzed literature.

Answer: Thank you for your observation. When we talked about manageability, we meant that all of factors could be divided into three groups – manageable, partly manageable or unmanageable. We classified factors in a particular group depending on whether they can be managed (and/or controlled) by the state or not; we made it on the base of analyzed literature and CCS projects’ background around the world. So, we didn’t use any statistical variables as the averages or medians.

Point 5. Line 267: How the Dpl value was evaluated?

Answer: Thanks for your question. We took this value from Table 2 (total score, last line).

Point 6. Figure 3: I suggest other method of correlation evaluation. Simple linear correlation calculation is unsufficient in my opinion. The model should be proved in more appropriated way.

Answer: Thank you for your comment. We carried out regression analysis (Line 295)

Point 7. Line 286: I have not found the Table 4. Do you mean Fig. 4? 

Answer: Thank you for your remark. We meant Table 2 (Assessment matrix), we corrected it.

Point 8. Line 328: In discussion I would add some references to other publiaction (if they have been published so far) treated about the problem.

Answer: Thank you for your recommendation. We added some references related to the problem. 

Point 9. General comment: The authors did not provided collected information about the number of project in several countries, which were analyzed. It could provide additional information.

Answer: Thank you for your comment, we added some references, where the information about the number of CCS projects were collected (table 3)

Round 2

Reviewer 1 Report

I have checked the revised manuscript.

The authors have answered my questions and concerns.

I recommend this revised manuscript to be published in Resources.

Best wishes.

Reviewer 2 Report

Thank you very much for your kind responses.